# Effects of Adding Blueberry Residue Powder and Extrusion Processing on Nutritional Components, Antioxidant Activity and Volatile Organic Compounds of Indica Rice Flour

**DOI:** 10.3390/biology11121817

**Published:** 2022-12-14

**Authors:** Xinzhen Zhang, Yang Gao, Ran Wang, Yue Sun, Xueling Li, Jin Liang

**Affiliations:** 1Key Laboratory of Jianghuai Agricultural Product Fine Processing and Resource Utilization of Ministry of Agriculture and Rural Affairs, Anhui Agricultural University, Hefei 230036, China; 2Anhui Engineering Laboratory for Agro-Products Processing, Anhui Agricultural University, Hefei 230036, China; 3College of Tea & Food Science and Technology, Anhui Agricultural University, Hefei 230036, China

**Keywords:** blueberry residue, indica rice flour, extrusion, antioxidant properties, volatile organic compounds

## Abstract

**Simple Summary:**

The aim of this study was to investigate the effects of adding blueberry residue and extrusion processing on nutritional components, antioxidant activity and volatile organic compounds of indica rice flour. The results showed that, after adding blueberry residue and extruding, the content of fat and total starch decreased significantly, while the content of total dietary fiber and anthocyanins increased significantly, and the effect of scavenging DPPH and ABTS+ free radicals was enhanced, and the antioxidant capacity in vitro was improved. In addition, GC—IMS analysis detected 104 volatile compounds in indica rice expanded powder containing blueberry residue and indica rice expanded powder, and 86 volatile organic compounds were successfully identified. The contents of 16 aldehydes, 17 esters, 10 ketones and 8 alcohols were significantly increased. Therefore, it is an efficient and innovative processing method to add blueberry residue powder to indica rice flour and extrude it, which can significantly improve its nutritional value, antioxidant properties and flavor substances, and has broad application prospects in the production of nutritious food and functional food.

**Abstract:**

Using indica rice flour as the main raw material and adding blueberry residue powder, the indica rice expanded powder (REP) containing blueberry residue was prepared by extrusion and comminution. The effects of extrusion processing on the nutritional components, color difference, antioxidant performance and volatile organic compounds (VOCs) of indica rice expanded powder with or without blueberry residue were compared. The results showed that the contents of fat and total starch decreased significantly after extrusion, while the contents of total dietary fiber increased relatively. Especially, the effect of DPPH and ABTS+ free radical scavenging of the indica rice expanded flour was significantly improved by adding blueberry residue powder. A total of 104 volatile compounds were detected in the indica rice expanded powder with blueberry residue (REPBR) by Electronic Nose and GC—IMS analysis. Meanwhile, 86 volatile organic compounds were successfully identified. In addition, the contents of 16 aldehydes, 17 esters, 10 ketones and 8 alcohols increased significantly. Therefore, adding blueberry residue powder to indica rice flour for extrusion is an efficient and innovative processing method, which can significantly improve its nutritional value, antioxidant performance and flavor substances.

## 1. Introduction

Broken rice is the main by-product of grain processing. Although the nutritional composition of broken rice is similar to that of whole rice [1], its processing and utilization forms are limited. The expanded blending powder can be used as the utilization form of broken rice processing. According to relevant reports, the physicochemical properties of extruded rice flour changed significantly after adding mung bean starch [2]. In addition, poria cocos was mixed with rice flour, and the nutrient powder was made by twin-screw extrusion, which greatly improved its sensory and mixing characteristics. The product had a unique flavor and showed good blending characteristics [3]. Furthermore, some functional ingredients, such as blueberry residue, maybe added to the expanded mixing powder for improving the product quality. Blueberry residue is the main by-product of blueberry fruit processing, containing functional components such as anthocyanins and other active substances [4,5], which need to be properly developed and utilized. However, according to our previous research, most of the dietary fiber contained in blueberry residue was insoluble dietary fiber, and soluble dietary fiber plays a major role in the human body. Therefore, it was particularly important to modify dietary fiber of the blueberry dregs. Extrusion processing could change the internal structure and properties of materials, break the connecting bonds between cellulose polymers, and thus achieve the purpose of transforming insoluble dietary fiber into soluble dietary fiber [6], which may improve its utilization.

In addition, extrusion processing can not only improve the flavor quality of food, but also effectively promote digestion and absorption. A relative report showed that co-extrusion of proanthocyanidins from bayberry leaves with rice flour can significantly reduce its digestibility and starch crystallinity [7]. Similarly, a study demonstrated that extrusion could be an effective processing technique to enhance the antioxidant activities of apple pomace powder, significantly increasing the antioxidant activity (ORAC) in the in vitro gastrointestinal digesta [8]. Therefore, it is possible to use the mixture of crushed indica rice flour and blueberry residue powder as raw materials to prepare the blending powder by extrusion processing. It could improve the quality of blended powder and promote the value-added utilization of blueberry residue and broken rice.

The rice products processed by extrusion and added with purple sweet potato could improve the structural characteristics, reduce the digestibility and improve the quality of rice products, which had been studied in our laboratory [9]. Therefore, on the basis of previous studies, the extrusion characteristics of rice flour were further studied by adding other functional components such as blueberry residue. In this study, the changes of nutritional components and antioxidant activity of indica rice flour with blueberry residue powder after extrusion were investigated. In addition, electronic nose and gas chromatography-ion mobility spectrometry (GC—IMS) were used to compare the changes of volatile organic compounds (VOCs) in different samples.

## 2. Materials and Methods

### 2.1. Materials

#### 2.1.1. Raw Material

Broken indica rice was purchased from Anhui Wanyaohu Food Co., Ltd., Tongling, China. with 7.16% protein, 1.17% dietary fiber, 0.88% fat and 0.4% ash. Blueberry residue was purchased from Anhui Duxiushan blueberry Technology Co., Ltd., Anqing, China. with 8.08% protein, 39.17% dietary fiber and 3.30% fat. All other chemicals and solvents were of analytical reagent grade. ABTS [2,2-diazo-di (3-ethyl-benzothiazole-6-sulfonic acid) diammonium salt] and DPPH [1,1-diphenyl-2-trinitrophenylhydrazine] were purchased from Aladdin reagent (Shanghai) Co., Ltd., Shanghai, China.

#### 2.1.2. Preparation and Treatment of Samples

Blueberry residue powder and indica rice flour were mixed evenly at a mass ratio of 12:88%. The water content was adjusted to 16%, and then put into the feed port of the twin-screw extruder (DSE32-1, Jinan Shengrun Extrusion Machinery Co., Ltd., Jinan, China). When the extrusion temperature was 50–80–110–130 °C, the feed rate was 13 Hz, and the screw rotation rate was 17 Hz, the obtained REPBR was dried in an oven at 65 °C for 3 h, and then it was put into a high-speed multi-function grinder (SS-1022, Yongkang Bo’ou Hardware Co., Ltd., Yongkang, China) for crushing, and then it was placed in a sealed bag for cooling and stored at 4 °C until further analysis.

### 2.2. Methods

#### 2.2.1. Determination of Nutritional Components

The nutritional components of the indica rice expanded powder (REP) and the indica rice expanded powder with blueberry residue (REPBR), including total starch, protein, fat and total dietary fiber, were determined according to the national standard. The total starch was determined by acid hydrolysis, protein by the Kjeldahl method, the fat by Soxhlet extraction, and the total dietary fiber by the enzyme gravimetric method. For the determination of anthocyanin content, refer to the reported method [10].

#### 2.2.2. Determination of Color Difference

Firstly, the white board and standard color card were used to calibrate the white balance and color of the color difference meter, and then the sample photos were taken on the special white board for measurement. A total of 5 points were taken to measure the L* value (brightness), a* value (red green), b* value (yellow green) of each point, and then to remove a maximum value and a minimum value. Finally, the average value ± standard deviation were taken [11].

#### 2.2.3. Determination of Antioxidant Capacity

The determination method of DPPH clearance was carried out as described above, with appropriate modifications [12,13]. The prepared DPPH was put into 0.06 mmol/L solution with anhydrous methanol, 1 mL of extract was taken and mixed with 4 mL of DPPH solution, and then the mixture was kept away from light at room temperature for 30 min. Anhydrous methanol was used as blank, the absorbance value was measured at 517 nm, and we tried to avoid light during the whole test. The calculation formula was as follows:(1)$$\mathrm{DPPH}\;\mathrm{clearance}\;\left(\%\right)=1-\frac{A_{2\;}-A_{1}}{A_{0}}\times 100$$
where: A_0_ blank: the absorbance value after the reaction of 1 mL of anhydrous methanol with 4 mL of DPPH solution; A_1_ control: absorbance value after reaction of 1 mL extract with 4 mL anhydrous methanol; A_2_ sample: absorbance value after reaction of 1 mL extract with 4 mL DPPH solution.

The determination method of ABTS+ clearance was carried out as described above, with appropriate modifications [14,15]. First, 5 mL of 7.4 mmol/L ABTS solution and 5 mL of 2.6 mmol/L potassium persulfate solution were mixed and placed in the dark for 15 h. Then the mixture was diluted with anhydrous methanol to the absorbance value of 0.70 ± 0.02. Under the condition of avoiding light, 0.2 mL of sample solution was sucked into the beaker, then 4 mL of ABTS mixture was added, and the absorbance value was measured at 734 nm after 6 min of reaction. The calculation formula was as follows:(2)$$\mathrm{ABTS}+\;\mathrm{clearance}\;\left(\%\right)=1-\frac{A_{2}-A_{1}}{A_{0}}\times 100$$
where: A_0_ blank: the absorbance value after the reaction of 0.2 mL of anhydrous methanol with 4 mL of ABTS+ solution; A_1_ control: absorbance value after reaction of 0.2 mL extract with 4 mL anhydrous methanol; A_2_ sample: absorbance value after reaction of 0.2 mL extract with 4 mL ABTS+ solution.

#### 2.2.4. Determination of Electronic Nose

The odor was detected by an electronic nose (PEN3, AIRSENSE, Schwerin, Germany). A total of 2 g of REP and REPBR was taken into a headspace bottle, the bottle mouth was sealed with the bottle cap, and was used after enrichment at 50 °C for 10 min, the headspace sampling method was used to conduct electronic nose analysis and detection on the two samples. The specific parameters of electronic nose detection were: head space pre injection time of 5 s, automatic zeroing time of 10 s, test time of 120 s, cleaning time of 120 s, sensor chamber flow of 300 mL/min, and initial injection flow of 300 mL/min [16].

#### 2.2.5. Volatile Organic Compounds (VOCs) by GC-IMS

The volatile organic compounds (VOCs) were determined by GC-IMS. For analysis, 2 g of sample powder was weighed and transferred to a 20 mL headspace flask and maintained at 80 °C and 500 rpm for 20 min. Then, headspace sampling (500 µL) was carried out with a heated syringe (65 μC) (Automatic injection syringe). A WAX capillary column (30 m × 0.53 mm ID, 1 µm) was used to separate the VOCs with a column temperature of 60 °C and a drift tube temperature of 45 °C. High purity N_2_ was used as the sample carrier gas, and the procedure flow was as follows: the initial concentration was 2 mL/min, maintained for 2 min, then increased to 10 mL/min in 10 min, and finally gradually increased to 100 mL/min in the remaining 20 min. All VOCs were analyzed in triplicate.

## 3. Statistical Analysis

The data were analyzed by IBM SPSS Statistics 26 and origin 2021. All data were expressed as mean ± standard deviation. One-way analysis of variance (ANOV A) followed the least significant difference (LSD), and multiple comparison tests were used to test the mean difference between the two groups. *p* < 0.05 was statistically significant.

## 4. Results and Discussion

### 4.1. Analysis of Nutritional Components

Blueberry residue contains 39.17 mg dietary fiber per 100 g, and the anthocyanin content is 4.54 mg C3G/g. As shown in Table 1, compared with REP, the content of total dietary fiber and anthocyanins in REPBR increased significantly after extrusion, It was reported that blueberry residue contain rich dietary fiber and anthocyanins [17]. Indica rice flour contains 76.15 g starch and 0.88 g fat per 100 g, while blueberry residue contains 3.30 g fat per 100 g. After extrusion, the fat and total starch content decreased significantly, which might be due to the formation of starch polysaccharide fatty polymer with the fat in raw materials during extrusion [18], or the degradation of starch molecules during extrusion, as well as the hydrolysis of some triglycerides into monoglycerides and free fatty acids [19,20].

### 4.2. Color Difference

The value of L* and b* of REPBR decreased significantly compared with REP and Mixed powder, however, the value of a* increased (Table 2). This might be caused by the degradation and conversion of anthocyanins in blueberry residue in the process of high temperature and high-pressure extrusion [21], or the Maillard reaction of REPBR in the process of high temperature and high-pressure extrusion, leading to its color deepening.

### 4.3. Antioxidant Capacity Analysis

Antioxidant activity was assessed by measuring the ability of antioxidants to scavenge free radicals. The commonly used methods were the DPPH method and the ABTS method [22]. Figure 1 showed that REP had a certain scavenging capacity for DPPH· and ABTS+, which might be due to the fact that the indica rice flour contains antioxidants such as V_E_ and V_B_ [23], and the Maillard reaction might occur in the extrusion process of the indica rice flour, and the Maillard reaction products had a certain antioxidant capacity [24]. Compared with REP, the optimized REPBR significantly improved the removal ability of DPPH· and ABTS+, it was reported that anthocyanins had a strong antioxidant capacity [25], while blueberry residue powder contained more anthocyanins.

### 4.4. Electronic Nose Analysis

The change in electronic nose signal value may be caused by the formation of aroma compounds with different concentrations [26]. Figure 2 showed that the REPBR had obvious response values in W1W, W1S, W5S, W2S and W2W sensors. W1W was sensitive to inorganic sulfide; W1S sensitive aroma was methyl; the sensitive aroma of W5S was nitrogen oxide; the sensitive aroma of W2S was alcohol ether aldehyde ketone; W2W was sensitive to aromatic components and organic sulfides [27].

This might be due to the high content of alcohols, aldehydes, ketones and hydrocarbons in the samples after extrusion processing after adding blueberry residue powder to indica rice flour. Alcohols and aldehydes were mainly obtained by oxidative decomposition of fat; esters were formed by an esterification reaction of alcohol and fatty acid during extrusion; ketones might be produced by alcohols’ high-temperature oxidation products and ester decomposition [28,29,30]. It might also be that REPBR causes starch gelatinization and degradation, protein denaturation and recombination, Maillard reaction of carbonyl amino compounds, and degradation of phenols and flavonoids under high temperature, high pressure and high shear extrusion to produce new flavor substances [31]. In order to further judge the influence of adding blueberry residue after extrusion on the composition of volatile organic compounds, qualitative and quantitative analysis should be carried out in combination with GC-IMS technology.

### 4.5. Analysis of Volatile Organic Compounds (VOCs) by GC—IMS

Flavor is one of the most important characteristics of rice processing. The changes in volatile components of indica rice flour during extrusion processing after adding blueberry residue powder were studied by GC—IMS. Figure 3A showed the three−dimensional map of REP and REPBR. It intuitively showed the differences of VOCs in different samples. However, due to the inconvenience of observation, the difference comparison was made in the vertical view, and the results were shown in Figure 3B. The background of the whole figure was blue. The ordinate represented the retention time (s) of gas chromatography, and the abscissa represented the ion migration time. The red vertical line at the abscissa 1.0 was the reactive ion peak (RIP, normalized), and each point on both sides of the rip peak represented a volatile organic compound. The substance concentration of VOCs could be expressed by color. The darker the color, the greater the concentration [32]. White indicates low concentration and red indicates high concentration. In addition, a difference comparison model was also carried out to illustrate the different VOCs between the REP and the REPBR, as shown in Figure 3C. The spectrum of the REPBR was obtained by deducting the reference spectral background (the REP was used as a reference). The blue dot area indicated that the substance was lower than that of the REP in this sample, as shown in area a in the figure. The red area indicates that the substance was more than the REP in this sample, as shown in area b in the figure. It was obvious that after the addition of blueberry residue to the indica rice flour and extrusion processing, the VOCs content had changed significantly, which could also be supported by the PCA results determined by the signal intensity of the VOCs (Figure 3D). It was observed that under the same experimental conditions, the difference between the REP and the REPBR in the distribution map was obvious and there was no overlap, which meant that the VOCs of the REP and the REPBR were significantly different.

In addition, the compounds were characterized by comparing the retention index and migration time data. As shown in Appendix A (which can be found in the Appendix A), a total of 104 volatile substances were observed in the REP and REPBR. Among them, 86 kinds of VOCs were successfully identified in the GC—IMS database. Some compounds produced different signals or spots (dimers or monomers) due to their different concentrations. The VOCs identified in the GC—IMS database belonged to several chemical families; specifically, there were 23 aldehydes, 3 acids, 11 alcohols, 26 esters, 2 ethers, 15 ketones, 2 heterocyclic compounds, 2 terpenoids, 1 aromatic aldehyde and 1 aromatic heterocyclic compound. This result was consistent with the previous study, which reported that alcohols, aldehydes, esters and ketones were the main volatile compounds of indica rice flour [33]. According to Appendix A, firstly, a higher content of alcohols could be observed in the REPBR, such as 1-hexanol, 1-octene-3-ol, pentanol, 2-methyl-1-butanol, 3-methyl-1-butanol, 1-pentene-3-ol, 2-methyl-1-propanol. Alcohol, such as 1-octene-3-ol, has the aroma of mushrooms and hay, while 1-hexanol can give the product a grass fragrance [34]. Secondly, the content of aldehydes (octanal, E-2-nonanal, E-2-octanal, E-2-heptaldehyde, E-2-hexanal, 2-valeraldehyde, heptaldehyde, hexanal, acrolein, 3-methylbutyraldehyde, 2-methylpropionaldehyde, propionaldehyde, acetaldehyde) after the addition of blueberry residue and extrusion processing was high. The aldehydes of C_8_~C_10_ had an obvious floral or fruity fragrance; for example, octanal has a strong orange aroma [35]. In addition, after extrusion (γ- Butyrolactone, ethyl E-2-hexanoate, ethyl heptanoate, ethyl butyrate, methyl 2-methylbutyrate, butyl acetate, butyl propionate, isobutyl acetate, ethyl 3-methylbutyrate, propyl acetate, ethyl acetate, methyl acetate, ethyl formate, methyl 2-methyl butyrate) and ketones (acetone, 1-hydroxy-2-acetone, 6-methyl-5-hepten-2-one, 2-pentanone, 2-pentanone, 2-butanone, 1-octen-3-one, 1-penten-3-one, 3-octen-2-one) also increased. The enrichment of ketones might be attributed to the high-temperature oxidation products of alcohols and the degradation of amino acids [36], while alcohols and fatty acids undergo esterification to form esters. However, the contents of ethyl 3-hydroxybutyrate, nonanal, hexyl 2-methylbutyrate, benzaldehyde, decanal, 2-heptanol, 1-butanol, cyclopentanone and other substances in the REP were higher. 

## 5. Conclusions

The nutritional components, color difference and antioxidant properties of the extruded powder after adding blueberry residue powder were studied, and the reliable information on the changes of different volatile organic compounds in the extrusion process of adding blueberry residue powder to indica rice flour was studied by electronic nose and GC—IMS for the first time. The results showed that the contents of fat and total starch decreased significantly, while the contents of protein and total dietary fiber increased significantly after extrusion processing. The ability of scavenging DPPH and ABTS+ free radicals was significantly improved by adding blueberry residue powder. A total of 104 volatile compounds were observed in REPBR, of which 86 were successfully identified by the GC—IMS database. The results showed that the acetic acid, γ- butyrolactone, e-2-octenal, e-2-pentenal, heptaldehyde, 3-methylbutyraldehyde, 2-methylpropionaldehyde, 3-methyl-1-butanol, methyl caproate, butyl acetate, butyl propionate, propyl acetate, methyl acetate, 1-penten-3-one, cyclohexanone, 2-pentylfuran, myrcene and other substances have a high content. Therefore, adding blueberry residue powder to indica rice flour and extruding was an efficient and innovative processing method, which could significantly improve its nutritional value, antioxidant properties and flavor substances.

## Figures and Tables

**Figure 1 biology-11-01817-f001:**
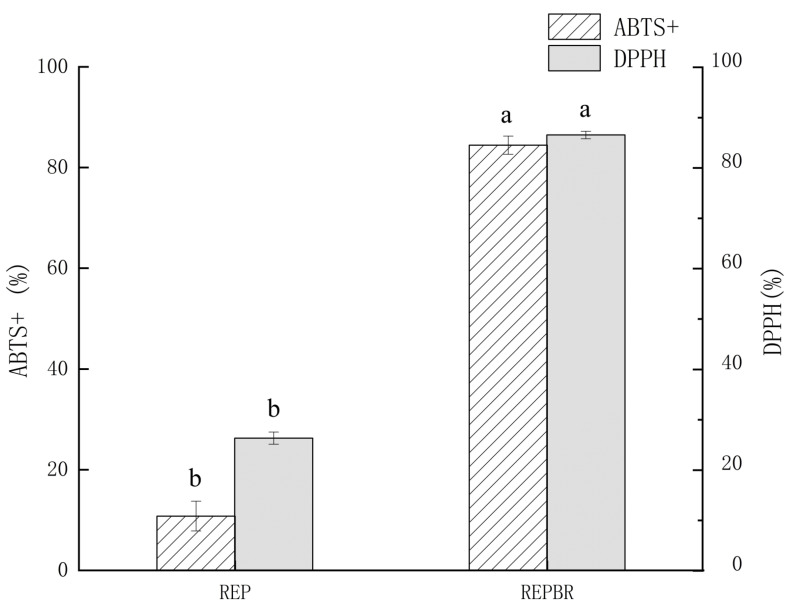
DPPH and ABTS+ radical scavenging efficiency of REP and REPBR. Different superscript letters are significantly different (*p < 0.05*).

**Figure 2 biology-11-01817-f002:**
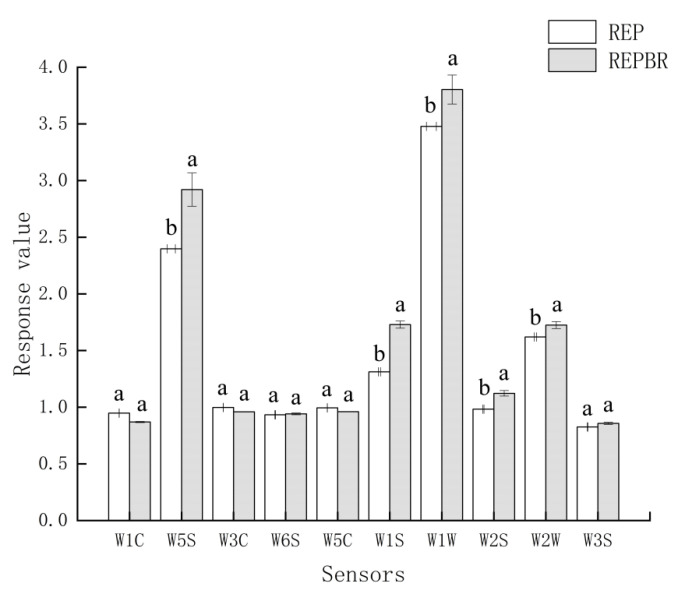
Electronic nose bar graph of REP and REPBR. Different superscript letters are significantly different (*p <* 0.05).

**Figure 3 biology-11-01817-f003:**
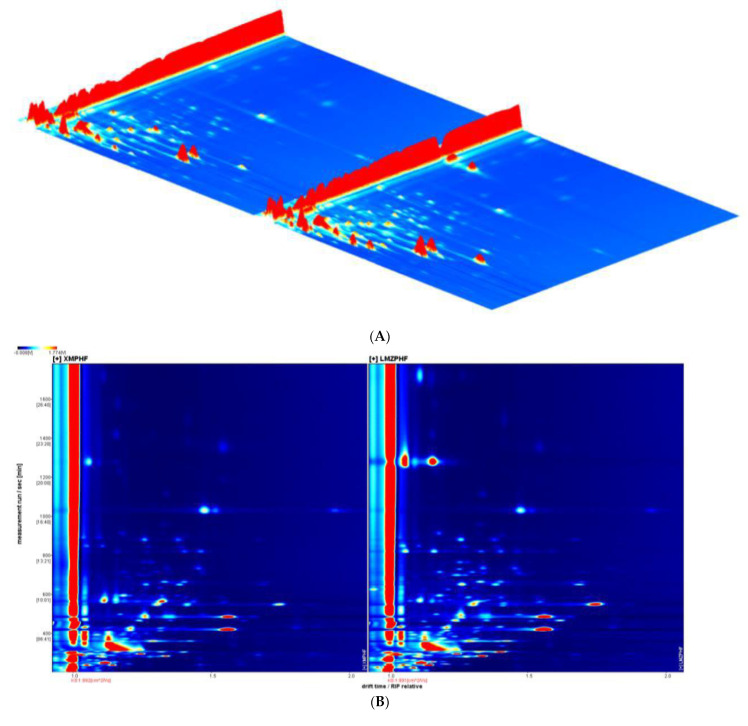
(**A**) GC—IMS three-dimensional spectra of REP and REPBR. (**B**) GC—IMS spectra of REP and REPBR (vertical view). (**C**) GC—IMS spectra of REP and REPBR (difference diagram). (**D**) PCA analysis diagram of REP and REPBR.

**Table 1 biology-11-01817-t001:** Nutritional composition of REP and REPBR.

	REP	REPBR
Total starch (g per 100 g)	74.80 ± 0.89 ^a^	68.50 ± 2.30 ^b^
Protein (g per 100 g)	6.72 ± 0.18 ^b^	7.91 ± 0.13 ^a^
Fat (g per 100 g)	0.56 ± 0.007 ^b^	0.11 ± 0.007 ^c^
Total dietary fiber (mg per 100 g)	1.03 ± 0.03 ^b^	2.28 ± 0.09 ^a^
Anthocyanins (mg C3G/g)	ND	0.14 ± 0.0005

Values are presented as mean ± standard deviation. Data in the same line with different superscript letters are significantly different (*p* < 0.05). ND, not detected.

**Table 2 biology-11-01817-t002:** Color difference between REP and REPBR.

	REP	REPBR	Indica Rice Flour	Blueberry Residue	Mixed Powder
L*	65.60 ± 0.42 ^b^	28.55 ± 1.07 ^d^	69.17 ± 2.25 ^a^	17.54 ± 0.26 ^e^	42.54 ± 0.32 ^c^
a*	−0.34 ± 0.057 ^e^	8.59 ± 0.35 ^b^	0.053 ± 0.0015 ^d^	12.51 ± 0.095 ^a^	7.35 ± 0.030 ^c^
b*	5.71 ± 0.30 ^a^	0.84 ± 0.055 ^d^	4.03 ± 0.18 ^b^	3.99 ± 0.010 ^b^	2.16 ± 0.012 ^c^

Values are presented as mean ± standard deviation. Data in the same line with different superscript letters are significantly different (*p* < 0.05).

## Data Availability

The data that has been used is confidential.

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
