# Peer review of "Effects of Adding Blueberry Residue Powder and Extrusion Processing on Nutritional Components, Antioxidant Activity and Volatile Organic Compounds of Indica Rice Flour"

_biology, 2022, doi:10.3390/biology11121817_

Round 1

Reviewer 1 Report

In the article “Effects of adding blueberry residue powder and extrusion processing on nutritional components, antioxidant activity and volatile organic compounds of indica rice flour” the authors prepared two extruded products and studied their chemical differences. The first product was made from rice, the second was made from rice mixed with blueberries. This experimental design, however, introduced major limitations. The authors repeatedly discuss that the differences between the samples are due to the changes during extrusion but do not consider the initial differences between the raw blends. Most importantly, the whole discussion part is lacking, especially if the abovementioned issue is skipped. The authors directly describe the results, the figures and how to look at them, but don’t actually discuss their meaning and implications, nor compare them to literature. The article needs language revision as it was difficult to follow and understand the text. Finally, the figures are poorly selected and presented. Taking all this into account I suggest rejecting this submission and kindly recommend the authors reassess this experiment and rewrite the article.

Here are my additional remarks:

The first half of the Introduction should be reorganized into a more clear and more logical structure. The first paragraph talks about multiple topics; it also mentions expansion before extrusion.

Line 72. Provide here more information about the extruder and/or give a reference.

Line 75. The Abstract mentions comminution, but it is not described in the Method.

Lines 70, 74, 79. The abbreviations were defined in the Abstract, but not in the article itself.

Line 86. What was the equipment used here?

Lines 113, 114. Instead of flavor (odor + taste), the electronic nose detected just odor or more specifically volatile compounds.

Lines 140-144. The differences in nutritional composition could be mostly explained by the initial difference in the nutritional composition of rice and blueberry. Please provide here the nutritional compositions of rice and blueberry powders (and their blend) before extrusion. Then compare these values with the values after extrusion in Table 1. Only after this, you can ascribe the differences between the two products to the extrusion process itself.

Lines 149-153. Same issue here, the colors of the initial raw blends were most probably significantly different because the blueberry powder by itself is dark (low L*) and red (high a*). It is questionable to attribute the changes in color after extrusion entirely to the transformation of anthocyanins/Maillard due to high temperature. The colors of the raw blends are needed here for comparison.

Line 172. You probably mean here “aroma compounds.”

Figure 2B. The three points are replicates? Specify what the ellipses are showing. Usually, ellipses are calculated 95% confidence regions, I think yours are drawn by hand.

Lines 186-195. Doing PCA for only two samples is pointless. Figure 2A already shows the differences between the two samples. Thus, roughly speaking, PC1 shows that the largest variation in the data comes from this difference (94% of it), and the remaining 6% is attributable to the variation between the replicates, which aligned in the PC2 direction. I think it is better to present Figure 2A in a different format in such a way that the values would be clearly visible e.g. a side-by-side barplot, or a lineplot. Then you can remove Fig 2B completely if you also add error bars or calculate statistics.

Lines 221-222. “VOCs content of blueberry residue powder changed”. You probably mean REP here.

The order of Figures 3A, 3B, and 3C is incorrect.

Figure 3D. A note: this PCA shows the same thing as Fig 2B. The difference between the two samples is greater than the difference between their replicates, meaning that the samples are statistically significantly different.

Figure 3D. Specify what the ellipses are showing. The three points are replicates?

Line 240. Specify that Table 4 can be found in Supplementary.

Figure 3E is too long and is not readable, it should be split and perhaps could be rotated.

Line 203-271. This part has a lot of mechanistic text describing the figures and how to read them but has no actual discussion of the results, their meaning, implications, or comparisons with the literature. What are the VOCs of the raw materials and how much did extrusion change them?

Author Response

Point 1:The first half of the Introduction should be reorganized into a more clear and more logical structure. The first paragraph talks about multiple topics; it also mentions expansion before extrusion.

Response 1: The introduction has been modified.

Point 2:Line 72. Provide here more information about the extruder and/or give a reference.

Response 2: Information about the extruder has been provided.

Point 3:Line 75. The Abstract mentions comminution, but it is not described in the Method.

Response 3: Comminution has been described in the method

Point 4:Lines 70, 74, 79. The abbreviations were defined in the Abstract, but not in the article itself.

Response 4: Abbreviations have been defined in the article.

Point 5:Line 86. What was the equipment used here?

Response 5: The equipment used has been described.

Point 6:Lines 113, 114. Instead of flavor (odor + taste), the electronic nose detected just odor or more specifically volatile compounds.

Response 6: It has been modified in the article.

Point 7:Lines 140-144. The differences in nutritional composition could be mostly explained by the initial difference in the nutritional composition of rice and blueberry. Please provide here the nutritional compositions of rice and blueberry powders (and their blend) before extrusion. Then compare these values with the values after extrusion in Table 1. Only after this, you can ascribe the differences between the two products to the extrusion process itself.

Response 7: Nutritional components of indica rice and blueberry dregs have been provided in the article

Point 8:Lines 149-153. Same issue here, the colors of the initial raw blends were most probably significantly different because the blueberry powder by itself is dark (low L*) and red (high a*). It is questionable to attribute the changes in color after extrusion entirely to the transformation of anthocyanins/Maillard due to high temperature. The colors of the raw blends are needed here for comparison.

Response 8:  It has been modified in the article.

Point 9:Line 172. You probably mean here “aroma compounds.”

Response 9:  It has been modified in the article.

Point 10:Figure 2B. The three points are replicates? Specify what the ellipses are showing. Usually, ellipses are calculated 95% confidence regions, I think yours are drawn by hand.

Response 10:  It has been modified in the article.

Point 11:Lines 186-195. Doing PCA for only two samples is pointless. Figure 2A already shows the differences between the two samples. Thus, roughly speaking, PC1 shows that the largest variation in the data comes from this difference (94% of it), and the remaining 6% is attributable to the variation between the replicates, which aligned in the PC2 direction. I think it is better to present Figure 2A in a different format in such a way that the values would be clearly visible e.g. a side-by-side barplot, or a lineplot. Then you can remove Fig 2B completely if you also add error bars or calculate statistics.

Response 11: Figure 2B has been deleted from the article as suggested

Point 12:Lines 221-222. “VOCs content of blueberry residue powder changed”. You probably mean REP here.

Response 12: In the article, it refers to REPBR.

Point 13:The order of Figures 3A, 3B, and 3C is incorrect.

Response 13: The order of the figures has been modified.

Point 14:Figure 3D. A note: this PCA shows the same thing as Fig 2B. The difference between the two samples is greater than the difference between their replicates, meaning that the samples are statistically significantly different.

Response 14: It has been modified in the article.

Point 15:Figure 3D. Specify what the ellipses are showing. The three points are replicates?

Response 15: The three points are obtained by repeating three experiments.

Point 16:Line 240. Specify that Table 4 can be found in Supplementary.

Response 16: It has been explained in the article.

Point 17:Figure 3E is too long and is not readable, it should be split and perhaps could be rotated.

Response 17: It has been modified in the article.

Point 18:Line 203-271. This part has a lot of mechanistic text describing the figures and how to read them but has no actual discussion of the results, their meaning, implications, or comparisons with the literature. What are the VOCs of the raw materials and how much did extrusion change them?

Response 18: The results have been explained and discussed in the article.

Reviewer 2 Report

In this manuscript, the authors compared the addition of blueberry residue powder to indica rice with or without extrusion, and characterized its effect on the nutritional components, color difference, antioxidant performance and volatile organic compounds. The methods they used are appropriate. However, the language is a bit hard to read. The authors should present the manuscript in a readable way.  

Author Response

Point 1:In this manuscript, the authors compared the addition of blueberry residue powder to indica rice with or without extrusion, and characterized its effect on the nutritional components, color difference, antioxidant performance and volatile organic compounds. The methods they used are appropriate. However, the language is a bit hard to read. The authors should present the manuscript in a readable way.  

Response 1:The article language has been modified.

Reviewer 3 Report

The paper should be formatted following Journal guidelines, including references, figures, and tables.

The Abstract should be rewritten following this structure: Introduction, Materials and Methods, Results, Discussion and Conclusion.

The novelty character of the paper should be better marked.

In Introduction, the advances state of research on natural products should be marked and related references such as:

Ramadan et al. Advances in Research on Food Bioactive Molecules and Health. Molecules. 2021 Dec 19;26(24):7678. doi: 10.3390/molecules26247678. 

A graphical scheme of study approach should be inserted.

Results in Figure 1 and 2 should be better described in the text.

Author Response

Point 1:The paper should be formatted following Journal guidelines, including references, figures, and tables.

Response 1:It has been modified according to the format of the journal guidelines.

Point 2:The Abstract should be rewritten following this structure: Introduction, Materials and Methods, Results, Discussion and Conclusion.

Response 2:It has been modified according to the article format.

Point 3:The novelty character of the paper should be better marked.

Response 3:The novelty of this study has been highlighted in the article.

Point 4:In Introduction, the advances state of research on natural products should be marked and related references such as:

Ramadan et al. Advances in Research on Food Bioactive Molecules and Health. Molecules. 2021 Dec 19;26(24):7678. doi: 10.3390/molecules26247678. 

Response 4:Relevant information on natural product research has been added to the introduction of the article.

Point 5:A graphical scheme of study approach should be inserted.

Response 5: The graphical scheme about the research method has been added in the supplement.

Point 6:Results in Figure 1 and 2 should be better described in the text.

Response 6: It has been modified in the article.

Round 2

Reviewer 1 Report

The revision has addressed only some of the issues but did not improve the article significantly and major issues are still present. Because a major methodological, content, and language revision is still needed, I recommend rejecting it.

Lines 13-16. None of the listed claims were investigated in the article.

Lines 37-42. “Natural products” are introduced, but their significance to the topic is not clear, as they are not discussed any further.

Line 58. It is not apparent from this text why any other method besides extrusion cannot be used.

Lines 282-292 describe the motivation for the study and should appear in Introduction instead.

Lines 162-163. This revision does not improve the major issue identified previously. Claims on lines 161-167 attributed to extrusion instead of just the initial differences in the raw materials should be substantiated by comparing extrudates with the raw materials before extrusion.

The same abovementioned major issue should be addressed on lines 173-177. What were the colors of the raw materials?

Lines 195-280. As commented in the first review, this text only describes how to read the figures and reiterates the results presented in the tables and figures but does not actually discuss them (meaning, implications, comparisons to literature, etc). For example, see the next point:

Lines 205-208. Starch gelatinization, protein denaturation, etc. Which of the listed processes could have produced the types of compounds detected by the electronic nose or GC-IMS? How would blueberry residue influence these processes and increase/decrease specific compounds?

Figure 2A. Spiderplot is generally a difficult to read type of plot and a simple side-by-side barplot or just a table would be more effective. http://perceptualedge.com/articles/dmreview/radar_graphs.pdf

Figure 3D. As commented in the first review, doing PCA with only two samples is pointless and this figure of PCA scores does not contribute anything to the article.

Figure 3E is still unreadable. Rotating and splitting it into multiple columns might help.

Table 2 in Supplementary. The statistical test should be done within columns, not rows.

This article requires language editing for grammar and style, including the revised parts.

Author Response

Point 1:Lines 13-16. None of the listed claims were investigated in the article.

Response 1: The simple summary has been modified.

Point 2:Lines 37-42. “Natural products” are introduced, but their significance to the topic is not clear, as they are not discussed any further.

Response 2: It has been modified in the article.

Point 3:Line 58. It is not apparent from this text why any other method besides extrusion cannot be used.

Response 3: The use of extrusion has been supplemented in the article.

Point 4:Lines 282-292 describe the motivation for the study and should appear in Introduction instead.

Response 4: Research motivation has been put in the introduction

Point 5:Lines 162-163. This revision does not improve the major issue identified previously. Claims on lines 161-167 attributed to extrusion instead of just the initial differences in the raw materials should be substantiated by comparing extrudates with the raw materials before extrusion.

Response 5: The extrudate has been compared with the raw material before extrusion.

Point 6:The same abovementioned major issue should be addressed on lines 173-177. What were the colors of the raw materials?

Response 6: It has been modified in the article.

Point 7:Lines 195-280. As commented in the first review, this text only describes how to read the figures and reiterates the results presented in the tables and figures but does not actually discuss them (meaning, implications, comparisons to literature, etc). For example, see the next point:

Response 7: It has been discussed in the article.

Point 8:Lines 205-208. Starch gelatinization, protein denaturation, etc. Which of the listed processes could have produced the types of compounds detected by the electronic nose or GC-IMS? How would blueberry residue influence these processes and increase/decrease specific compounds?

Response 8: It has been modified in the article.

Point 9:Figure 2A. Spiderplot is generally a difficult to read type of plot and a simple side-by-side barplot or just a table would be more effective. http://perceptualedge.com/articles/dmreview/radar_graphs.pdf

Response 9: It has been modified in the article.

Point 10:Figure 3D. As commented in the first review, doing PCA with only two samples is pointless and this figure of PCA scores does not contribute anything to the article

Response 10: It has been modified in the article.

Point 11:Figure 3E is still unreadable. Rotating and splitting it into multiple columns might help.

Response 11: The Figure 3E has been rotated in the article

Point 12:Table 2 in Supplementary. The statistical test should be done within columns, not rows.

Response 12: It has been modified in the article.

Round 3

Reviewer 1 Report

The authors added some discussion points and comparisons with the raw materials, as was requested by the previous reviews. This improved the article, though the overall quality remains low.

Lines 298-304 are duplicating the text in Introduction lines 52-57, thus do not add anything to the discussion.

Table 2 in supplementary. "Data in the same line with" -> "Data in the same column with"

Author Response

Point 1:Lines 298-304 are duplicating the text in Introduction lines 52-57, thus do not add anything to the discussion.

Response 1: It has been modified in the article.

Point 2: in supplementary. "Data in the same line with" -> "Data in the same column with"

Response 1: The table has been modified in the supplement.